# Sexually transmitted infection knowledge among men who have sex with men in Nairobi, Kenya

Delvin Kwamboka Nyasani[1,2]*, Onyambu Meshack Ondora[1], Laura Lusike Lunani[2], Geoffrey Oino Ombati[2], Elizabeth Mueni Mutisya[2], Gaundensia Nzembi Mutua[3], Matt. A. Price[3,4], Justus Osano Osero[1]

**1** Kenyatta University Department of Community Health and Epidemiology, Nairobi, Kenya, **2** KAVI Institute of Clinical Research, University of Nairobi, Nairobi, Kenya, **3** IAVI, New York, New York, United States of America, **4** Department of Epidemiology and Biostatistics, University of California, San Francisco, San Francisco, California, United States of America

* dnyasani@kaviuon.org, dnyasani01@yahoo.com

## Abstract

### Background

High rates of sexually transmitted infections (STIs) among men who have sex with men (MSM) have been reported, but there is little research on their STI knowledge. Our study sought to determine participants' characteristics that contribute to either high or low STI knowledge among MSM in Nairobi, Kenya.

### Methods

We mobilized MSM aged ≥18 years from Nairobi into a cross-sectional study. To determine their understanding of STIs, a pre-tested structured questionnaire was administered. Knowledge score was generated by summing up the number of responses answered correctly by a participant. We dichotomized scores as "low" and "high", by splitting the group at <12 and ≥12 which was the mean.

### Results

A total of 404 participants were interviewed between March and August 2020. The mean age was 25.2 (SD = 6.4) years. Majority were single (80.4%) and Christians (84.2%). All participants had some formal education ranging from primary to tertiary; the majority (92.3%) had secondary education or more. Most (64.0%) were employed and their monthly income ranged from <50->150 USD. Almost all (98.5%) were Kenyans. Of the 404 (90.6%) self-identified as male and (47.5%) reported to be exclusively top partners. Many (39.9%) reported being versatile, while those reporting to be bottom partners were, (12.6%). The last 12 months, (55.4%) of the participants reported having sex with men only and (88.6%) reported to have had multiple sexual partners. Participants scored an average of 12.2, SD 4.5. Multivariable backward elimination logistic regression revealed that participants who had tertiary education (aOR = 0.50, 95% CI 0.32–0.77), a higher income (aOR = 0.40, 95%

**Editor:** Hamid Sharifi, HIV/STI Surveillance Research Center and WHO Collaborating Center for HIV Surveillance, Institute for Future Studies in Health, Kerman University of Medical Sciences, ISLAMIC REPUBLIC OF IRAN

**Data Availability Statement:** All relevant data are within the manuscript and its Supporting Information files.The dataset can be accessed through the following link[https://www.openicpsr. org/openicpsr/project/192503/version/V1/view].

**Funding:** This work was partially funded by IAVI with the generous support of USAID and other donors; a full list of IAVI donors is available at www.iavi.org. The funders had no role in study design, data collection and analysis, decision to publish, or preparation of the manuscript.

**Competing interests:** The authors have declared that no competing interests exist.

CI 0.22–0.75) and were engaging in vaginal sex (aOR = 1.86, 95% CI 1.25–2.78) predicted significantly higher odds of high knowledge in the final multivariable model.

## Conclusion

Participant's knowledge level regarding STIs was low. We recommend health care workers to continue educating patients about STIs.

## Introduction

### Background

Sexually transmitted infections (STIs) are infections that are primarily transmitted through sexual contact. These infections can be caused by bacteria, viruses, parasites, or fungi and do affect both men and women. STIs can be spread during vaginal, anal, and oral sex. However, STIs can also be transmitted non-sexually through blood transfusions, sharing contaminated needles, and infected mother to her child during pregnancy, childbirth, or breastfeeding. The ease of transmission and possible health complications makes them a significant public health concern, as sexual activity is common among the population [1].

Over one million sexually transmitted infections (STIs) are acquired daily globally; common among them are chlamydia, gonorrhoea, syphilis and trichomoniasis. Many of these infections are frequently mild or asymptomatic [1]. In low- and middle-income countries (LMICs), symptomatic STIs are often treated by syndromic management which frequently misses asymptomatic STIs that therefore remain untreated and potentially infectious [2]. Regardless of symptoms, STIs can cause serious morbidity including cancer, infertility, and enhanced HIV transmission [3, 4].

The impact of STI epidemics is of particular importance among key, higher-risk populations including young people and adolescents, men who have sex with men (MSM), people in prisons, sex workers and people who inject drugs. These populations are often vulnerable to both HIV and other STIs not only because of their high-risk sexual and drug use behaviours, but also by structural barriers such as low access to quality health services, stigma and discrimination [5]. However, the risk may even be greater among MSM because unprotected receptive anal sex; is a greater risk factor for STI and HIV acquisition compared to unprotected vaginal sex [6].

For instance, several studies conducted in both developed and developing countries have shown that MSM exhibits a higher prevalence of STI compared to the general population [7–13]. A similar trend is also observed in Kenya, as indicated by [9]. Among the estimated 22,000 MSM in Kenya, the HIV prevalence was 18.2% compared to the national general population at 5.9%. These findings emphasize the need for targeted interventions aimed at addressing the issue of limited STI knowledge among MSM. However, it is worth noting that several studies have revealed a similar lack of STI knowledge not only among MSM but also within the general population [14–20].

Acquiring knowledge about symptoms and prevention of STIs is key in reducing the risk of contracting STIs among both the general and the key population [21, 22]. Being that the HIV prevalence among MSM in Kenya is estimated to be three times higher than that of the general population, we sought to characterize the socio-demographic and sexual behavioural characteristics that contribute to either high or low STI knowledge among MSM in Nairobi, Kenya.

## Materials and methods

### Study design

This was a cross-sectional study among sexually active MSM within Nairobi, Kenya, conducted between the months of March-August 2020. We interviewed participants to determine their understanding of STIs and assessed for association with socio-demographic characteristic and sexual behaviour.

### Study participants

The study was designed to enroll men aged ≥18 years and who have sex with men from places where MSM are known to either solicit their clients or where sexual activity was common including "cruising" areas along streets, or bars with or without lodging [23]. They had to be willing and able to give a written informed consent to participate. MSM who were intoxicated were excluded from participation.

### Participant recruitment

Mobilization was done by MSM community mobilizers. We identified MSM community mobilizers from the Nairobi-based Sex Workers Outreach Programme (SWOP) with whom we had partnered for a recent study [24]. The community mobilizers gave a brief description of the research activities to potential participants. Then using a systematic random sampling procedure they selected participants with the desired characteristics. Systematic random sampling is a sampling technique that the researcher selects participants to be included in the sample based on a systematic rule, using a fixed interval.

We had twenty one days to enrol 422 participants. We had purposively selected 5 sub-counties that were densely populated with MSM [23]. To get the percentage of the participants to be accessed from each sub-county, the researcher took the expected number in a specific sub-county divided by the total participants from all the 5 sub-counties and multiplied by 100. Then to get the numbers to be recruited from each sub county, it was the percentage of the participants to be accessed from each sub-county multiplied by the required number of participants (n = 422) (Table 1).

To get the first participant from each sub-county on each day of mobilization, we had 2 sealed papers written: "Yes and No". The mobilizers issued the first two participants the sealed papers if, for example, the second participant selected "Yes", then they were issued with a unique identification card. Then they continued mobilizing the second participant until the required number was attained. If the sampled participant had declined to participate, the next exiting participant was sampled. The sampled participants were then referred to the researcher and the research assistants for interviews at a selected venue. The selected venue was a safe and

**Table 1. The number of MSM in the selected sub-counties, at selected MSM geographic areas in Nairobi, Kenya and the sampled population.**

| Sub-County | Estimated number of MSM available on all selected MSM geographic areas | Percentage of the total MSM to be accessed | Number of MSM required for the study ((n = 422) |
|---|---|---|---|
| Starehe | 2000 (40 MSM geographic areas) | 55% | 232 |
| Westlands | 1000 (27 MSM geographic areas) | 27% | 114 |
| Dagoretti South | 360 (20 MSM geographic areas) | 10% | 42 |
| Ruaraka | 180 (15 MSM geographic areas) | 5% | 21 |
| Kasarani | 100 (10 MSM geographic areas) | 3% | 13 |
| Total | 3640 | 100 | 422 |

private place identified by the mobilizers where the interviews could be conducted; without attracting attention of the participants. At the entrance to the selected venue, a peer navigator confirmed participant identity by confirming that they had a unique identification card and that they were not intoxicated. Thereafter, written informed consent was obtained from the participants before the questionnaire was administered. Mobilization from the MSM geographic areas was done on 4 selected peak days (i.e., Wednesday, Friday, Saturday and Sunday) as identified by the key population implementing partners during the key population size estimate mapping [23]. During that time also the peak time identified was 18.00hrs-22.00hrs and that is what we adopted. Peak day refers to a day when the number of MSM present in the MSM geographic areas was more than usual.

## Procedure

Participants were assigned a unique study identification number. Data was collected about socio-demographics and sexual behaviour including age, marital status, religion, education level, employment status, level of income, nationality, gender identity, roles during sexual activity, sex with whom in the past 12 months, total number of sexual partners in the past 12 months, type of sex, if received gifts/money in exchange for sex, whether gave gifts/money in exchange for sex and history of having ever contracted any STI.

To determine their understanding about STIs participants were then given a pre-tested structured questionnaire (S1 Appendix). The questionnaire had four open ended questions and they had multiple correct responses (Table 4). Also, there was one short answer question. The total number of possible correct responses was 25. Interviewers were instructed not to prompt participants but only tally all correct responses provided. Items were scored as correct (score = 1) and incorrect (score = 0). If a participant gave an incorrect answer it was ignored (I.e., participants were not penalized for wrong answers). If a participant provided an answer deemed correct but not among the initial options, it was captured as other and later analysed. The lead researcher in the field did quality control checks on the questionnaires before the participant was reimbursed, allowing the team to avoid missing data or clarify ambiguities.

## Study variables

**Dependent variable.**   Knowledge score was the dependent variable. In assessing knowledge, the participants were asked the following questions; which diseases are spread via sexual intercourse? What the signs and symptoms of STIs were? How one could be infected with STIs? Can someone be infected with STIs via anal sexual intercourse? And through which media could one get infected with STIs? (Additional details are shown in the S1 Appendix). Interviewers were instructed not to prompt participants but only tally all responses provided. Correct answers were assigned a score of 1. Incorrect answers were not scored (i.e., given a score of 0,).

A knowledge score was generated by summing up the number of responses answered correctly by a participant (Table 4); to give a score ranging from 0 to 25. For analysis, we dichotomized scores as "low" and "high", by splitting the group at <12 and ≥12 (additional details are shown in S1 Appendix).

**Independent variables.**   Independent variables included in this analysis included age, marital status(single never married, married and single ever married), religion (Christians and non-Christians), education (primary, secondary and post-secondary), employment status (unemployed and employed), level of income (No income, <5000, 5001–10,000, 10001–15000 and >15000), nationality (Kenyans and non-Kenyans), gender identity (male, transgender woman, intersex and non-conforming), roles during sexual activity (top, bottom or versatile),

**Table 2. Socio-demographic characteristics of study participants and their association with participant's knowledge score.**

| Characteristics | Total (n = 404) | Knowledge score | |
|---|---|---|---|
| | | <12 (n = 215) | ≥12 (n = 189) |
| | n(%) | n(%) | n(%) |
| **Mean age = 25.20, SD = 6.41** | | | |
| **Age bracket** | | | |
| 18–24 | 241(59.7) | 127(52.7) | 114(47.3) |
| ≥25 | 163(40.3) | 73(44.8) | 90(55.2) |
| **Marital Status** | | | |
| Single never married | 325(80.4) | 162(49.8) | 163(50.2) |
| Married | 54(13.4) | 27(50.0) | 27(50.0) |
| Single ever married | 25(6.2) | 11(44.0) | 14(56.0) |
| **Religion** | | | |
| Christian(Catholics & protestants) | 340(84.2) | 174(51.2) | 166(48.8) |
| Others (Muslims, Atheists, Hindus) | 64(15.8) | 26(40.6) | 38(59.4) |
| **Education level** | | | |
| Primary | 31(7.7) | 19(61.3) | 12(38.7) |
| Secondary | 226(55.9) | 126(55.8) | 100(44.2) |
| Post-secondary | 147(36.4) | 55(37.4) | 92(62.6) |
| **Employment status** | | | |
| Unemployment/Student | 145(36.0) | 85(58.6) | 60(41.4) |
| Employment/Self-employed | 259(64.0) | 115(44.4) | 144(55.6) |
| **Level of income (KES)** | | | |
| None | 120(29.7) | 75(62.5) | 45(37.5) |
| <5000 | 54(13.4) | 28(51.9) | 26(48.1) |
| 5001–10000 | 87(21.5) | 38 (43.7) | 49(56.3) |
| 10001–15000 | 71(17.6) | 33(46.5) | 38(53.5) |
| >15000 | 72(17.8) | 26(36.1) | 46(63.9) |
| **Nationality** | | | |
| Kenyans | 398(98.5) | 196(49.2) | 202(50.8) |
| Non-Kenyans | 6(1.5) | 4(66.7) | 2(33.3) |

sex with whom in the past 12 months (men or both men and women), total number of sexual partners in the past 12 months (one or more than one), type of sex (vaginal, anal, oral, mutual masturbation), received gifts/money in exchange for sex (yes, all the time, yes, sometimes, No), gave gifts/money in exchange for sex (yes, all the time, yes, sometimes, No), and ever contracted STI (Yes, No) (see Tables 2 and 3).

## Data analysis

Data were entered into statistical package for the social sciences programme (IBM-SPSS) version 25. Quantitative data from the study questionnaires was coded. Single data entry was done into SPPS database after quality control checks had been done. Further cleaning was carried out after data entry using frequency distributions and cross tabulations until no more errors were detected. Descriptive statistics of socio-demographic and sexual behavioural variables were done. The relationship between each independent variable with dependent variable was determined using the bi-variable logistic regression analysis. Further, the multivariable logistic regression model included all variables with a p-value < 0.20 from the bivariate logistic regression analysis. Backward elimination was done on multivariable logistic regression model

**Table 3. Sexual behavioural characteristics of study participants and their association with participant's knowledge score.**

| Characteristics | Total (n = 404) | Knowledge score | |
|---|---|---|---|
| | | <12 (n = 215) | ≥12 (n = 189) |
| | n(%) | n(%) | n(%) |
| **Gender** | | | |
| Male | 366(90.6) | 176(48.1) | 190(51.9) |
| Transgender women | 13(3.2) | 8(61.5) | 5(38.5) |
| Intersex | 17(4.2) | 11(64.7) | 6(35.3) |
| Non-conforming | 8(2.0) | 5(62.5) | 3(37.5) |
| **Roles during sexual activity** | | | |
| Top(Insertive) | 192(47.5) | 94(49.0) | 98(51.0) |
| Bottom(Receptive) | 51(12.6) | 31(60.8) | 20(39.2) |
| Versatile (Both insertive & receptive) | 161(39.9) | 75(46.6) | 86(53.4) |
| **Sex with whom in the past 12 months** | | | |
| Men | 224(55.4) | 123(54.9) | 101(45.1) |
| Both men & women | 180(44.6) | 77(42.8) | 103(57.2) |
| **Total number of sexual partners in the past 12 months** | | | |
| Only 1 | 46(11.4) | 30(65.2) | 16(34.8) |
| >1 | 358(88.6) | 170(47.5) | 188(52.5) |
| **Type of sex** | | | |
| Vaginal | 170(42.1) | 69(72.4) | 101(27.6) |
| Anal | 378(93.6) | 185(48.9) | 193(51.1) |
| Oral | 115(28.5) | 50(43.5) | 65(56.5) |
| Mutual masturbation | 79(19.6) | 34(43.0) | 45(57.0) |
| **Received gifts/money in exchange for sex** | | | |
| Yes, all the time | 38(9.4) | 14(36.8) | 24(63.2) |
| Yes, sometimes | 192(47.5) | 93(48.4) | 99(51.6) |
| No | 174(43.1) | 93(53.4) | 81(46.6) |
| **Gave gifts/money in exchange for sex** | | | |
| Yes, all the time | 16(4.0) | 5(31.3) | 11(68.8) |
| Yes, sometimes | 139(34.4) | 71(51.1) | 68(48.9) |
| No | 249(61.6) | 124(49.8) | 125(50.2) |
| **Ever contracted STI** | | | |
| Yes | 175(43.3) | 81(46.3) | 94(53.7) |
| No | 229(56.7) | 119(52.0) | 110(48.0) |

to identify the final significant variables and adjusted odds ratios (AOR) with 95% confidence interval (CI) were calculated. A statistically significant result was considered when the p-value was <0.05 and was deemed statistically significant. The detailed outcomes were presented in tables.

## Ethical consideration and study approval number

Approval of the study was sought from the Kenyatta University Board of Post- Graduate Studies. Ethical Clearance was given by the Kenyatta University ethics review committee; (PKU/1071/11121). Permission to conduct the study was sought from the National Council for Science, Technology and Innovations (NACOSTI), and Nairobi County and sub-county facilities. All participants were informed about the purpose of the study prior to becoming involved in the study. Those who agreed to participate in the study gave a written informed

consent. Confidentiality and anonymity of the information given by the participants was protected by ensuring that the names of the participants were not indicated in the data collection tools. Participants were reimbursed in the local currency equivalent of $2 in compensation for their time.

## Results

### Demographics

A total of 404 participants were interviewed between March and August 2020. They had a mean age of 25.2 (SD = 6.4) years. The majority were single never married (80.4%; 325) and Christians (84.2%; 340). All participants had some formal education ranging from primary to tertiary level; the majority (92.3%; 373) had secondary education or more. Most of the participants (64.0%; 259) were employed and their level of monthly income ranged from $<50->150 USD. Almost all the participants (98.5%; 398) were Kenyans. (Table 2)

### Sexual behaviour

Most of the participants (90.6%; 366) self-identified as male and almost half (47.5%; 192) of them reported to be exclusively insertive ("Top") partners. Many (39.9%; 161) reported being versatile (both bottom and top), while those reporting to be receptive ("Bottom") partners were, (12.6%; 51). In the last 12 months, (55.4%; 224) of the participants reported having sex with men only and (44.6%; 180) reported being bisexual. Also during that period, majority of the participants (88.6%; 358) reported having multiple sexual partners (Table 3).

The majority of the respondents (93.6%; 378) reported having anal sex, (42.1%; 170) had vaginal sex, (28.5%; 115) reported oral and (19.6%; 79) reported mutual masturbation. Almost half (47.5%; 192) of the participants, reported receiving gifts/money in exchange for sex, sometimes. The majority of the participants (95.3%; 385), engaged in transactional sex. Participants who reported to have ever contracted STI were (43.3%; 175). (Table 3)

### STI knowledge

Out of a total of 25 possible correct answers, we observed an average score of 12.2 correct, SD 4.5 (additional details shown in Appendix 1).As indicated in Table 4 below, majority of the participants were aware of gonorrhoea (92.8%; 375), syphilis (88.1%; 356), HIV (65.6%; 265), genital warts (33.9%; 137), Herpes Simplex (22.8%; 92), chancroid (20.5%;83), hepatitis B (17.3%;70), chlamydia (13.6%;55) and trichomoniasis (5.2%;21).

Regarding signs and symptoms of STIs, most of the participants were cognizant of penile discharge (66.8%; 270), burning sensation during urination (60.4%; 244), genital ulcer/sores (39.6%; 160), anal discharge (21.0%; 85), swelling in the groin region (23.0%; 93), anal ulcer/sore (25.0%; 101), anal pain (21.3%; 86), and scrotal swelling (14.1%; 57).

Also almost all the participants knew that STIs were transmitted through unprotected sexual intercourse (97.5%; 394) i.e. vaginal, anal, oral and mutual masturbation. Also about the anal sexual intercourse risk to STIs, majority of the participants (83.7%; 338) were aware that they could be infected with STIs through unprotected anal sexual intercourse. About three quarters of the participants reported correctly that they could be infected with STIs through semen (78.0%; 315) and through blood (46.0%; 186), vaginal fluids (32.7%; 132), contaminated sharp objects (17.3%; 70) and anal fluid (29.2%; 118).

**Table 4. Proportion of participants answering STI knowledge items correctly.**

| | Proportion Correct | Score |
|---|---|---|
| **Sexually Transmitted Infections** | | |
| Gonorrhoea | 92.8 | 1 |
| Syphilis | 88.1 | 1 |
| HIV | 65.6 | 1 |
| Genital warts | 33.9 | 1 |
| Herpes simplex | 22.8 | 1 |
| Chancroid | 20.5 | 1 |
| Hepatitis B | 17.3 | 1 |
| Chlamydia | 13.6 | 1 |
| Trichomoniasis | 5.2 | 1 |
| **Symptomatology of STIs** | | |
| Penile discharge | 66.8 | 1 |
| Burning pain during urination | 60.4 | 1 |
| Genital ulcer/sores | 39.6 | 1 |
| Anal ulcer/sores | 25.0 | 1 |
| Swelling in groin region | 23.0 | 1 |
| Anal pain | 21.3 | 1 |
| Anal discharge | 21.0 | 1 |
| Scrotal swelling | 14.1 | 1 |
| **Transmission mechanism** | | |
| Unprotected sexual intercourse | 97.5 | 1 |
| Contaminated sharp objects/wounds | 5.9 | 1 |
| Blood transfusion | 3.0 | 1 |
| Unhygienic conditions | 2.2 | 0 |
| Don't know | 1.0 | 0 |
| **Anal intercourse exposure to STI** | 83.7 | 1 |
| **Transmission media** | | |
| Semen | 78.0 | 1 |
| Blood | 46.0 | 1 |
| Vaginal fluids | 32.7 | 1 |
| Anal fluids | 29.2 | 1 |
| Contaminated sharp objects | 17.3 | 0 |
| Clothing | 7.2 | 0 |
| Food | 1.7 | 0 |

\*\*Points tallied for each answer given to provide a composite "knowledge score"

**N/B**- you can access the questionnaire through the following link: [https://www.openicpsr.org/openicpsr/project/192503/version/V1/view]

## Participant's socio-demographic factors associated with their STI knowledge score

Table 5 presents the results of the bivariable analysis, which examines the associations between various socio-demographic characteristics and STI knowledge among the participants. The study revealed a statically significant association between STI knowledge and the participant's education level, employment status and level of income. Specifically, participants with tertiary education (post- secondary) were found to be 2.65 times more likely to have a higher STI

**Table 5. Bivariable logistic regression of participant's socio-demographic characteristics independently associated with their STI knowledge score.**

| Variable | Knowledge Score>12 | P-Value | Unadjusted OR (95% CI) |
|---|---|---|---|
| **Age group** | | | |
| 18-24yrs (Reference) | 114/241 (47.3%) | | |
| ≥25yrs | 90/163(55.2%) | 0.12 | 1.37 (0.92–2.05) |
| **Marital status** | | 0.85 | |
| Single (Reference) | 163/325 (50.2%) | | |
| Married | 27/54 (50.0%) | 0.57 | 0.79 (0.35–1.79) |
| Single ever married | 14/25 (56.0%) | 0.62 | 0.79 (0.30–2.04) |
| **Religion** | | | |
| Catholics & protestants (Reference) | 166/340 (48.8%) | | |
| Muslim, Atheists and Hindu | 38/64 (59.4%) | 0.12 | 1.53 (0.89–2.64) |
| **Education level** | | | |
| "Up to completed Primary (Reference) | 12/31(38.7%) | 0.001 | |
| "Up to completed Secondary | 100/226(44.2%) | 0.56 | 1.26 (0.58–2.711) |
| "Up to completed Post-Secondary | 92/147(62.6) | 0.02 | 2.65 (1.20–5.87) |
| **Employment status** | | | |
| (Not-employed) (Reference) | 60/145 (41.4%) | | |
| Employed | 144/259 (55.6%) | 0.006 | 1.77 (1.18–2.68) |
| **Level of income (KES)** | | | |
| (No income) Reference | 45/120 (37.5%) | 0.006 | |
| <5000 | 26/54 (48.1%) | 0.187 | 1.55 (0.81–2.96) |
| 5000–10000 | 49/87(56.3%) | 0.008 | 2.15 (1.23–3.77) |
| 10001–15000 | 38/71(53.5%) | 0.032 | 1.92 (1.06–3.48) |
| >15000 | 46/72(63.9%) | 0 | 2.95 (1.61–5.41) |
| **Nationality** | | | |
| Kenyans (Reference) | 202/398 (50.8%) | | |
| Non-Kenyans | 2/6 (33.3%) | 0.407 | 0.49 (0.09–5.41) |

knowledge score compared with the participants who had primary education (Crude OR: 2.65, 95% CI 1.20–5.87). Similarly, employed participants were 1.77 times more likely to have a higher STI knowledge score compared with the participants who were not employed (Crude OR: 1.77, 95% CI 1.18–2.68). Furthermore, participants who had a higher level of income were 2.95 times more likely to have a higher STI knowledge score compared with the participants who were not earning (Crude OR: 2.95, 95% CI 1.61–5.41). Conversely, the study did not find any statically significant association between STI knowledge score and the participant's age, marital status, religion and nationality (Table 4).

## Multivariable logistic regression of participant's socio-demographic characteristics independently associated with their knowledge about STIs

In the multivariable logistic regression model, we included all predictor variables with P-values < 0.2 from the bivariable logistic regression analysis. To determine the final significant variables, we used the backward elimination selection technique. The variables considered for multivariable analysis was age, religion, education status, employment status, and level of income. However, among these variables, only education status and level of income remained to be significantly associated with the knowledge score (aOR = 0.50, 95% CI 0.32–0.77) and (aOR = 0.40, 95% CI 0.22–0.75) respectively (Table 6).

**Table 6. Multivariable logistic regression of participant's socio-demographic characteristics independently associated with their knowledge about STIs.**

| Variable | Adjusted OR 95% CI | P-Value |
|---|---|---|
| **Education level** | | |
| "Up to completed Primary" (Reference) | | 0.004 |
| "Up to completed Secondary | 0.38 (0.17–0.87) | 0.022 |
| "Up to completed Post-Secondary | 0.50 (0.32–0.77) | 0.002 |
| **Level of income (KES)** | | |
| No income (Reference) | | 0.016 |
| <5000 | 0.40 (0.22–0.75) | 0.004 |
| 5000–10000 | 0.70 (0.33–1.48) | 0.349 |
| 10001–15000 | 0.94(0.48–1.83) | 0.854 |
| >15000 | 0.76 (0.38–1.52) | 0.441 |

## Participant's sexual behaviour characteristics associated with their knowledge score

Table 7 presents the results of the bivariable analysis, which examined the associations between sexual behaviour and STI knowledge among the participants. The study revealed a statically significant association between STI knowledge and the participant's sexual behaviour i.e. sex with whom, total number of sexual partners and the type of sex (vaginal).

Specifically, participants who were bisexual they were 1.63 times more likely to have a higher knowledge score compared with the participants who were gay (Crude OR: 1.63, 95% CI 1.10–2.42). Additionally, Participants who had multiple sexual partners were 2.07 times more knowledgeable compared to the ones who had one sexual partner (Crude OR: 2.07, 95% CI 1.09–3.94). Also, Participants who had vaginal sexual intercourse were 1.86 times more likely to have a higher knowledge score compared with the participants who were not (Crude OR: 1.86, 95% CI 1.25–2.78). On the other hand, the study did not find any statically significant association between STI knowledge score and the participant's gender, participant role during sexual activity, type of sex (anal, oral & mutual masturbation), and transactional sex and ever contracted STI (Table 7).

## Multivariable logistic regression of participant's sexual behavioural characteristics independently associated with their knowledge about STIs

In the multivariable logistic regression model, we included predictor variables with P-values < 0.2 from the bivariable logistic regression analysis. To determine the final significant variables, we utilized the backward elimination selection technique. The variables considered for multivariable analysis were gender, roles during sexual activity, sex with whom, total number of sexual partners, type of sex (vaginal& oral), and transactional sex. Among these variables, only the type of sex (vaginal sex) remained to be significantly associated with participant's knowledge score (aOR = 1.86, 95% CI 1.25–2.78) (Table 8).

## Discussion

STI knowledge and awareness is a key element towards decreasing the incidence of STIs among populations at high risk of infection. However, findings from this study among MSM in Nairobi, Kenya, demonstrated that half of them had low STI knowledge.

Participants most commonly mentioned gonorrhoea, syphilis and HIV. This could be attributed to the symptomatic nature of gonorrhoea and syphilis, whereas for HIV it might be

**Table 7. Bivariable logistic regression of participant's sexual behavioural characteristics independently associated with their STI knowledge score.**

| Variable | Knowledge Score>12 | Unadjusted OR 95% CI | P-Value |
|---|---|---|---|
| **Gender** | | | |
| Male (Reference) | 190/366 (51.9%) | | 0.38 |
| Transgender women | 5/13 (38.5%) | 0.58 (0.19–1.80) | 0.35 |
| Intersex | 6/17 (35.3%) | 0.51 (0.18–1.41) | 0.19 |
| Non-conforming | 3/8 (37.5%), | 0.56 (0.13–2.36) | 0.43 |
| **Roles during sexual activity** | | | |
| Top(insertive) Reference | 98/192 (51.0%) | | 0.21 |
| Bottom (Receptive) | 20/51 (39.2%) | 0.62(0.33–1.16) | 0.14 |
| Versatile (Both insertive & receptive | 86/161(53.4%) | 1.10(0.72–1.67) | 0.66 |
| **Sex with whom** | | | |
| Men only (Reference) | 101/224 (45.1%) | | |
| Both Men & women | 103/180 (57.2) | 1.63(1.10–2.42) | 0.02 |
| **Total number of sexual partners** | | | |
| Only one (Reference) | 16/46 (34.8%) | | |
| >1 | 188/358 (52.5%) | 2.07(1.09–3.94) | 0.03 |
| **Type of sex** | | | |
| Vaginal intercourse | 101/170 (27.6%) | 1.86(1.25–2.78) | 0.002 |
| Anal | 193/378 (51.1%) | 1.42 (0.64–3.18) | 0.39 |
| Oral | 65/115 (56.5%) | 1.40(0.91–2.17) | 0.13 |
| Mutual masturbation | 45/79(57.0%) | 1.38(0.84–2.27) | 0.20 |
| **Received money/gifts in exchange for sex** | | | |
| Yes, all the time (Reference) | 24/38 (63.2%) | | 0.17 |
| Yes, sometime | 99/192(51.6%) | 0.62(0.30–1.27) | 0.19 |
| No | 81/174 (46.6%) | 0.51(0.25–1.05) | 0.07 |
| **Gave money/gifts in exchange for sex** | | | |
| Yes, all the time (Reference) | 11/16 (68.8%) | | 0.34 |
| Yes, sometime | 68/139 (48.9%) | 0.44(0.14–1.32) | 0.14 |
| No | 125/249 (50.2%) | 0.46(0.16–1.36) | 0.16 |
| **Ever contracted STI** | | | |
| Yes (Reference) | 94/175 (53.7%) | | |
| No | 110/229(48.0%) | 0.80(0.54–1.18) | 0.26 |

due to the extensive and vigorous awareness campaigns conducted both nationally and globally. On the other hand, relatively a few participants reported other STIs such as genital warts, herpes simplex, chancroid, hepatitis B, chlamydia and trichomoniasis. This knowledge did not correlate well with prevalent STIs in the region. In coastal Kenya, MSM engaging in receptive anal intercourse (RAI) had an estimated prevalence of 21.2% for rectal chlamydia and gonorrhoea infections, with an incidence rate of 53.0 per 100 person -years [25]. In Nairobi, the prevalence of rectal gonorrhoea and chlamydia among MSM practicing rectal anal intercourse (RAI) was found to be 5.6% and 3.2%, respectively, for non-sex worker MSM, while male sex

**Table 8. Multivariable logistic regression of participant's sexual behavioural characteristics independently associated with their knowledge about STIs.**

| Variable | P-Value | Adjusted OR 95%CI |
|---|---|---|
| **Type of sex** | | |
| Vaginal intercourse | 0.002 | 1.86 (1.25–2.78) |

workers had a prevalence of 5.0% and 4.3% [10]. Another study conducted in Western Kenya among high-risk populations reported that the MSM population had the highest prevalence of Hepatitis B at 17.4% [26]. Moreover, in Mombasa the prevalence of anogenital warts among HIV- uninfected and infected MSM was found to be 2.9% and 9.4% respectively [27]. Additionally, a study conducted among tertiary student men who have sex with men in Nairobi reported prevalence rates of 58.8% for chlamydia, 51.1% for gonorrhoea, 1.5% for trichomoniasis, and 0.7% for latent syphilis [28].

Based on our study, when we compare the reported STI prevalence reported by different authors to the participant's knowledge, we have observed that the relationship between MSM knowledge and STI prevalence varies. For example, with some STIs participants exhibited high knowledge, but this did not always match the reported the prevalence of those infections (e.g., gonorrhoea). Hence, it is important to provide STI health education to the participants regardless of the current prevalence rates; as some of the STIs such as Herpes Simplex Virus type 2 (HSV-2) and Syphilis infections put participants at a higher risk of acquiring HIV infection [3, 4]

Regarding the signs and symptoms of STIs, more than half of the participants knew about penile discharge and burning sensations during urination. The other signs and symptoms namely: genital ulcer/sores, swelling in the groin region, anal discharge, anal ulcers/sores, anal pain and scrotal swelling were mentioned by a few participants. Similarly, other studies have also reported low knowledge among MSM [14, 16]. However, worth noting is that the measures of STI knowledge used on those studies do vary from those used in our survey.

This study highlights a significant challenge in relying exclusively on participants to seek treatment for STIs; particularly considering that some infections may be asymptomatic as noted by [25, 29, 30]. Therefore, it is crucial to prioritize health education for the participants to increase their awareness and knowledge about STIs. As research has shown that informed patients are more likely to seek healthcare services compared to those who lack knowledge [31]. Seeking timely treatment is essential as it can prevent the transmission of STIs to others. To address asymptomatic STIs, it is recommended to implement regular STI screenings every 3–6 months for MSM at high risk of contracting STIs [32]). By implementing such measures, we can make advancements in tackling the spread of STIs and promoting overall health among this population.

All the participants had reported of engaging in anal sexual intercourse and they understood the potential risk of contracting STIs. However, according to a study by [33], it was found that individuals who practiced anal sex were unlikely to seek STI screening services. Therefore, there is a need to raise awareness among the participants about the importance of undergoing regular STI screenings.

In this study, some participants revealed their involvement in oral sexual intercourse. As a result, it is vital to conduct regular screenings at least annually for extra-genital areas such as the rectum and pharynx among MSM [34]. This is to ensure that extra-genital STIs are not left undetected.

This study did not reveal a statistically significant association between knowledge of the MSM and their age. However, contrary to our findings a study that was conducted among MSM in Ireland reported that participants who were aged 18–24 years of age had lower knowledge about STIs compared with the older MSM [35]. Low knowledge about STIs among the youth could be occasioned by government policies and laws that criminalize key population behaviours and by education and health systems that pay no attention to or reject them[5].

The research findings from our study revealed that participants who had tertiary education had a higher knowledge score compared with the participants who had primary education. Consistent with our findings is a study which was conducted by [36] in Melaka Malaysia; they

found that participants who had tertiary education were found to be twice as likely to be knowledgeable about STIs compared to those without such education.

Further, this study revealed that participants with income they had higher STI knowledge levels compared to those without any income. This observation is consistent with [37] who highlighted the close connection between an individual's socioeconomic status as determined by factors like income and education and access to STI information. Additionally, this study revealed that participants who engaged in vaginal sex (bisexual men) they had higher STI knowledge score compared to gay men. These findings are contrary to [38, 39] who reported that bisexual men had limited availability of culturally sensitive education materials or health information that is specifically targeted to their needs. However, their settings are different from our Kenyan context.

In our study, participants had low knowledge levels and also the majority had multiple sexual partners. Interestingly, our findings contrast with a previous study conducted among MSM in the UK, which revealed that individuals who possessed a higher level of STI knowledge had multiple sexual partners. The authors of that study suggested that besides knowledge, various psychological and eco-social factors played a crucial role in influencing behaviours [20].

A similar observation was also noted in a study conducted in Estonia, involving 772 subjects, which aligns with the findings of the UK study. The study revealed that higher knowledge scores did not correlate with lower rates of HIV infection[40]. This finding affirms that although knowledge plays a significant role in behaviour change models, it is often insufficient on its own to drive actual changes in behaviour. Models like the Information-Behavioural Skills Model suggest that individuals require not only knowledge about STIs but also motivation to prevent them and the necessary skills to implement risk reduction measures, such as regular STI testing [14].

## Limitations

Our study has limitations. Our sample represents those men who seek sex at selected geographic areas in Nairobi, and may not be representative of MSM, but otherwise do not visit these areas. However, we aimed to enroll very high at -risk individuals. Additionally, our data are cross sectional, and as such we cannot infer causality between our independent variables and our dependent variable i.e. knowledge of STIs. Also, we did not test for STIs and thus we are not able to make any comparison with knowledge of STIs and prevalence of STIs in this study population.

## Conclusion

The participants in our study demonstrated a low level of knowledge regarding STIs. While it has been acknowledged in previous research that reducing the risk of STI infection requires more than just knowledge about STIs, we still strongly recommend HCWs to continue in educating patients about these infections. This is crucial because patients who are enlightened are more likely to seek healthcare services, thereby playing a vital role in preventing the further spread of STIs to their sexual partners.

## Implications to programmes

Programmatic implications arise from the findings of this study. It is evident that individual health education and psychological approaches are effective in equipping individuals with knowledge and skills to adopt new behaviours aimed at reducing risks [41]. However, in our study, participants demonstrated low knowledge about STIs, coupled with a significant proportion engaging in multiple sexual partnerships.

To address these issues, we strongly recommend HCWs to continue providing health talks about STIs; specifically tailored for MSM during their visits to healthcare facilities and in other relevant forums. It is important to note that knowledge alone may not be sufficient to initiate behaviour change, as supported by other studies. Therefore, active follow-up of high-risk individuals is necessary to prevent STIs.

Effective strategies for follow-up can include recalling MSM for regular check-ups, particularly those who report engaging in high-risk sexual behaviours. This can be facilitated through phone calls, engaging peer leaders, or community health workers. Additionally, it is crucial to create demand for the STI prevention cascade. Also utilizing prompts incorporated within electronic MSM records can enable the implementation of timely and consistent reminders, such as bulk short message reminders, for offering STI testing to high-risk MSM.

Considering the likelihood of extra-genital STIs not being detected by both patients and clinicians, it is imperative to conduct at least annual screening of extra-genital sites, such as the rectum and pharynx, among sexually active MSM. This approach will help in detecting and treating extra-genital STIs that may otherwise go unnoticed and contribute to the spread of infections.

Furthermore, due to the limitations of syndromic management of STIs, which may result in the missed diagnosis of asymptomatic STIs, we recommend implementing quarterly STI screenings for individuals at risk. This proactive approach will enable the detection and treatment of asymptomatic STIs, thus effectively curbing the transmission of STIs.

## Implications to policymakers

Policymakers should prioritize the development of clear and comprehensive policies and guidelines to assist HCWs in responding effectively when an individual self-identifies as MSM. These policies should establish a framework for connecting MSM individuals with appropriate programs or healthcare facilities that can provide regular screening and treatment for STIs.

Additionally, policymakers should consider the implementation of self-testing options for STIs. Self-testing offers several advantages, including convenience, privacy, and the ability for patients to avoid potential embarrassment while still accessing necessary STI care. In this regard, policymakers can explore the utilization of self-sampling kits for various STI tests. As previous studies have shown that HIV self-testing was well-received among similar populations, indicating the potential acceptance of self-sampling kits for STI testing among MSM individuals [42]. By incorporating these measures into policies, policymakers can facilitate better healthcare outcomes for MSM individuals and promote their overall well-being.

## Supporting information

**S1 Appendix. Supplemental questionnaire results: Indicating participant's knowledge score.**
(PDF)

## Acknowledgments

We would like to acknowledge the participants and the research assistants who assisted in data collection. We also appreciate Kenneth Ekoru from the imperial university and Janet Muasya from the University of Nairobi for their input in the data analysis.

## Author Contributions

**Conceptualization:** Delvin Kwamboka Nyasani, Onyambu Meshack Ondora, Matt. A. Price, Justus Osano Osero.

**Data curation:** Delvin Kwamboka Nyasani, Laura Lusike Lunani, Geoffrey Oino Ombati.

**Formal analysis:** Delvin Kwamboka Nyasani, Laura Lusike Lunani, Geoffrey Oino Ombati, Elizabeth Mueni Mutisya, Gaundensia Nzembi Mutua, Matt. A. Price, Justus Osano Osero.

**Funding acquisition:** Delvin Kwamboka Nyasani.

**Investigation:** Delvin Kwamboka Nyasani.

**Methodology:** Delvin Kwamboka Nyasani.

**Project administration:** Delvin Kwamboka Nyasani, Justus Osano Osero.

**Resources:** Delvin Kwamboka Nyasani, Matt. A. Price.

**Supervision:** Onyambu Meshack Ondora, Justus Osano Osero.

**Validation:** Onyambu Meshack Ondora, Laura Lusike Lunani, Geoffrey Oino Ombati, Elizabeth Mueni Mutisya, Gaundensia Nzembi Mutua, Matt. A. Price, Justus Osano Osero.

**Writing – original draft:** Delvin Kwamboka Nyasani.

**Writing – review & editing:** Delvin Kwamboka Nyasani, Onyambu Meshack Ondora, Laura Lusike Lunani, Geoffrey Oino Ombati, Elizabeth Mueni Mutisya, Gaundensia Nzembi Mutua, Matt. A. Price, Justus Osano Osero.

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
