## [Decision Letter · Decision Letter 0]

1 Mar 2023

PONE-D-23-02660Sexually transmitted infection knowledge levels, socio-demographic characteristics and sexual behaviour among men who have sex with men:  results from a cross-sectional survey in Nairobi, Kenya.PLOS ONE

Dear Dr. Nyasani,

Thank you for submitting your manuscript to PLOS ONE. After careful consideration, we feel that it has merit but does not fully meet PLOS ONE’s publication criteria as it currently stands. Therefore, we invite you to submit a revised version of the manuscript that addresses the points raised during the review process.

We look forward to receiving your revised manuscript.

Kind regards,

Hamid Sharifi

Academic Editor

PLOS ONE

“This work was partially funded by IAVI with the generous support of USAID and other donors; a full list of IAVI donors is available at www.iavi.org.  The contents of this manuscript are the responsibility of IAVI and co-authors and do not necessarily reflect the views of USAID or the US Government. We would like to acknowledge the participants and the research assistants who assisted in data collection. We also appreciate Kenneth Ekoru from the imperial university and Janet Muasya from the University of Nairobi for their input in the data analysis.”

“DKN received a scholarship to study her Masters degree. Conception and design of the study was done by DKN.This work was partially funded by IAVI with the generous support of USAID and other donors; a full list of IAVI donors is available at www.iavi.org.  The funders had no role in study design, data collection and analysis, decision to publish, or preparation of the manuscript”

Reviewers' comments:

Reviewer's Responses to Questions

**Comments to the Author**

1. Is the manuscript technically sound, and do the data support the conclusions?

Reviewer #1: Yes

Reviewer #2: Partly

Reviewer #3: Yes

2. Has the statistical analysis been performed appropriately and rigorously? 

Reviewer #1: Yes

Reviewer #2: Yes

Reviewer #3: No

3. Have the authors made all data underlying the findings in their manuscript fully available?

Reviewer #1: Yes

Reviewer #2: Yes

Reviewer #3: Yes

4. Is the manuscript presented in an intelligible fashion and written in standard English?

Reviewer #1: Yes

Reviewer #2: Yes

Reviewer #3: Yes

5. Review Comments to the Author

Reviewer #1: The conclusion in the manuscript is supported by data from the results section. The sample size is adequate for an academic study.

Statistical analysis was done as expected where after identifying crude measures of association the author went on to find independent risk factors through multivariable analysis.

It has been submitted in standard English with no grammatical errors.

Reviewer #2: Ombati et al. report the level of understanding of sexually transmitted infections (STIs) in men who have sex with men (MSM). The sociodemographic characteristics and specific knowledge of STIs of the participants are well presented. The study's main outcome is met; however, conclusions should not be made from non-significant findings. This cross-sectional study is welcome and needed as it considers a topic ostracized in many African communities like Kenya's. The discussion also approaches the sexual behaviour of MSM in Nairobi (e.g. multiple partners and oral sexual intercourse) and puts it into perspective with other populations and health interventions to consider. One can also appreciate the good knowledge of some STIs reported, highlighting Kenya's progress in educating the at-risk population, which can still be improved.

Line 54 – numbers 1 to 10 are usually written in scientific writing (e.g. one million instead of 1 million).

The paper is easy to read, although it could benefit from an attentive punctuation revision. For instance, in lines 56 and 77 – Add a dot to mark the end of the sentence; in line 335 – add a dot in 1026.

In the methodology, it could be clearer how the knowledge score ranging from 0 to 29 was given based on the 5 questions.

Results/Abstract – There is no significance on the participants aged 25 and older being more likely to have a higher knowledge score than the younger ones (odds ratio touches 1.0). Same for the type of employment and those earning/not earning. Hence, no significant differences should be mentioned in the abstract ("participants aged ≥25 years were more likely to have a higher knowledge score compared with the participants aged 18-24 years (aOR=0.973, CI: 0.616-1.538").

It could be interesting to highlight in the discussion which STIs could benefit more from the health education of the target population — perhaps by relating the knowledge of the study population of each STI included in the questionnaire and the prevalence/burden of that disease in men who have sex with men in Kenya or Nairobi.

Reviewer #3: 1-The multivariable modelling revealed that participants who were aged ≥25 years were more

323 likely to have a higher knowledge score compared with the participants who were aged 18-24

324 years (adjusted odds ratio aOR=0.973, 95% CI 0.616-1.538)

This interpretation is not correct as 95% CI 0.616-1.538 include 1

2- Regarding occupation participants who were employed had a higher

328 knowledge score compared to the ones who were not employed (aOR=0.922, 95% CI 0.401-

329 2.117)

this is not correct for same reasen

3 -Under level of income, participants earning Kshs 5000-10,000 were three times likely to have a higher knowledge score compared to the ones who were not earning (aOR 2.332, 95% CI 0.990-6.263) . Participants who were earning Kshs >15000 (USD >150) were also three times more likely to have a higher knowledge score compared to the ones who were not earning (aOR=2.520, 95% CI 0.900-7.055). not correct

4- (aOR= 1.550, 95% CI 1026-2.342). Please provide detail of IC calculation

6. PLOS authors have the option to publish the peer review history of their article (what does this mean?). If published, this will include your full peer review and any attached files.

Reviewer #1: **Yes: **Charles Uzande

Reviewer #2: **Yes: **Luís-Jorge Amaral

Reviewer #3: **Yes: **GUY FRANCK BIAOU ALE

---

## [Author Response · Author response to Decision Letter 0]

6 Jul 2023

UNIVERSITY OF NAIROBI

KAVI-INSTITUTE OF CLINICAL RESEARCH

Telephone ; 2717694, 2714613, 2725404

Mob : 0722 207 417 or 0734 333 143

Fax:2727703

Kenyatta National Hospital

P.O. Box 19676-00202

Nairobi, Kenya

Delvin Nyasani BSN 05 Jul 2023

Research Nurse

KAVI-Institute of Clinical Research

University of Nairobi

+254 720898197 (mobile)

Email: dnyasani@kaviuon.org

Hamid Sharifi

Academic Editor

Public Library of Science (PLOS ONE)

Dear Editor,

We are delighted that PLOS ONE will consider publication of our paper pending satisfactory revisions as suggested by the editor and reviewers. 

We have given careful consideration to all the editor’s and reviewers’ comments and have done our best to address them all. The following is a point by point explanation of how we have addressed the concerns and revised our manuscript. The line number(s) on the revised manuscript with Track Changes and text highlight has been specified to show the text representing each response. 

Editor’s comments

Response

We have adhered to the PLOS ONE template

Response

Thank you, all participants gave a written informed consent. Please see line 217 -218 on the sub title of ethical considerations and study approval. 

“This work was partially funded by IAVI with the generous support of USAID and other donors; a full list of IAVI donors is available at www.iavi.org. The contents of this manuscript are the responsibility of IAVI and co-authors and do not necessarily reflect the views of USAID or the US Government. We would like to acknowledge the participants and the research assistants who assisted in data collection. We also appreciate Kenneth Ekoru from the imperial university and Janet Muasya from the University of Nairobi for their input in the data analysis.”

“DKN received a scholarship to study her Masters degree. Conception and design of the study was done by DKN. This work was partially funded by IAVI with the generous support of USAID and other donors; a full list of IAVI donors is available at www.iavi.org. The funders had no role in study design, data collection and analysis, decision to publish, or preparation of the manuscript”

Response

Thank you, we have deleted the funding statement from the acknowledge section. The funding statement should remain the way it is.

Response

Thank you, we have uploaded the data. To view log into https://www.openicpsr.org/openicpsr/project/192503/version/V1/view

Response

We have deleted and also added some references. 

Reviewers' comments:

5. Review Comments to the Author

Reviewer #1: The conclusion in the manuscript is supported by data from the results section. The sample size is adequate for an academic study.

Statistical analysis was done as expected where after identifying crude measures of association the author went on to find independent risk factors through multivariable analysis.

It has been submitted in standard English with no grammatical errors.

Reviewer #2: Ombati et al. report the level of understanding of sexually transmitted infections (STIs) in men who have sex with men (MSM). The sociodemographic characteristics and specific knowledge of STIs of the participants are well presented. The study's main outcome is met; however, conclusions should not be made from non-significant findings. This cross-sectional study is welcome and needed as it considers a topic ostracized in many African communities like Kenya's. The discussion also approaches the sexual behaviour of MSM in Nairobi (e.g. multiple partners and oral sexual intercourse) and puts it into perspective with other populations and health interventions to consider. One can also appreciate the good knowledge of some STIs reported, highlighting Kenya's progress in educating the at-risk population, which can still be improved.

Line 54 – numbers 1 to 10 are usually written in scientific writing (e.g. one million instead of 1 million). 

Response

We have revised as indicated on the second paragraph of the sub-title background line 62.We also appreciate your comment about not over interpreting non-significant results, and have revised our conclusions to reflect this.

The paper is easy to read, although it could benefit from an attentive punctuation revision. For instance, in lines 56 and 77 – Add a dot to mark the end of the sentence; in line 335 – add a dot in 1026.

Response

We have revised. Please see line 64, Line 77 we have revised the whole paragraph. 

Line 335 revised now its line 330before Table 5.

In the methodology, it could be clearer how the knowledge score ranging from 0 to 29 was given based on the 5 questions.

Response

Correction has been done the knowledge score is ranging from 0-25 and also we have revised table 4 to include a column with correct response/score. See also line 162- 166 on the second paragraph of the procedure sub-title.

Results/Abstract – There is no significance on the participants aged 25 and older being more likely to have a higher knowledge score than the younger ones (odds ratio touches 1.0). Same for the type of employment and those earning/not earning. Hence, no significant differences should be mentioned in the abstract ("participants aged ≥25 years were more likely to have a higher knowledge score compared with the participants aged 18-24 years (aOR=0.973, CI: 0.616-1.538").

Response

We have revised our abstract and results so that non-significant results are not reported in such a way as to be interpreted as significant or otherwise important.

It could be interesting to highlight in the discussion which STIs could benefit more from the health education of the target population — perhaps by relating the knowledge of the study population of each STI included in the questionnaire and the prevalence/burden of that disease in men who have sex with men in Kenya or Nairobi.

Response

We have revisited this early in our discussion on the second paragraph, highlighting the burden of STIs among MSM in various regions in Kenya and we are relating it to STI knowledge level findings among MSM in our study. Please see lines 342- 357.

Reviewer #3: 1-The multivariable modelling revealed that participants who were aged ≥25 years were more

323 likely to have a higher knowledge score compared with the participants who were aged 18-24

324 years (adjusted odds ratio aOR=0.973, 95% CI 0.616-1.538)

This interpretation is not correct as 95% CI 0.616-1.538 include 1

2- Regarding occupation participants who were employed had a higher

328 knowledge score compared to the ones who were not employed (aOR=0.922, 95% CI 0.401-

329 2.117)

this is not correct for same reason-

Response

 We have revised and included only the statistically significant values i.e. the level of education and bisexual men.

3 -Under level of income, participants earning Kshs 5000-10,000 were three times likely to have a higher knowledge score compared to the ones who were not earning (aOR 2.332, 95% CI 0.990-6.263) . Participants who were earning Kshs >15000 (USD >150) were also three times more likely to have a higher knowledge score compared to the ones who were not earning (aOR=2.520, 95% CI 0.900-7.055). not correct

4- (aOR= 1.550, 95% CI 1026-2.342). Please provide detail of IC calculation

Response

We used the SPSS software for our multivariable logistic regression, the confidence intervals were provided in the output.

Summary of Reviewer

I. The author reports low STI knowledge among MSM from a sample in Kenya.

II. It was important to report specific knowledge gaps on STIs as the majority (from the proportions reported line 314 Table 4) of MSM was knowledgeable about gonorrhea, syphilis and HIV/AIDs which constitute the syndromic approach in STI diagnosis and treatment. 

Response

We do acknowledge your comment that the participants were knowledgeable about gonorrhea, syphilis and HIV/AIDs which constitute the syndromic approach in STI diagnosis and treatment. We have indicated the burden of STIs among the MSM Line 342-357. We do propose that because other STIs are asymptomatic, regular STI screening to be implemented. See line 372- 381

One wouldn't expect anyone non-medical to know about specific STIs such as chancroid unless common terms like Bubo, urethral discharge, warts e.t.c are used.

Response

We acknowledge your suggestions. In regards to the listed STIs, we did find that some participants were familiar with them. However, it is worth noting that for certain STIs, participants used colloquial terms instead. For instance, some individuals were not familiar with the term "gonorrhea," but they referred to it as the "burning disease" or "kuchomeka" in their local slang and we recorded it as gonorrhea. Similarly, Hepatitis B they referred to it as the STI that causes yellow eyes. This was also recorded as a “correct” response (for Hepatitis)

III. After a multivariable analysis the only variable that was statistically significant was tertiary education Line 326 and 327 ( aOR=2.627, 327 95% CI 1.142-6.043) Confidence Interval does not include 1. Otherwise, all other variables were not significant. 

Response

We have revisited our results to highlight only those variables that significantly correlated with our knowledge score, which included education and bisexuality. Please see line 326-330.

IV. However, the author can still report all other variables from the univariate analysis.

Response

Thank you for your comment. We focused on the multivariable analysis.

Major Issues

There were no major issues in the study.

Minor Issues

I. The literature review done is biased towards comparing the prevalence of STI among MSM and the general population instead of knowledge levels among the two groups this can be corrected.

Response

We have revised this. Please see line 82-84.

II. The study reports the use of mobilizers who were already engaged by the organization and the assumption is that they were already doing some community work including working with the MSMs that were also enrolled in the study. It is important to highlight how these clients were managed in the study as they were likely to introduce information bias.

Response

We acknowledge the likelihood of information bias in the data recorded from the participants. However, information bias was either minimized or eliminated through training and all the mobilizers were blinded to the outcome of interest.

The study used a set of questions to assess knowledge levels among MSM it was going to provide more evidence if tools such as the STD KQs (Matthew Lee Smith 2020) among others were also referred to.

Response

We do acknowledge that we did not employ the STD KQs tool to assess the participant’s knowledge and instead used a tool generated by the team at our research institution. The tool that we used was reviewed by my co-authors and it was pretested for comprehension before it was used. 

We thank the editor and reviewers for their comments. With these revisions, we feel the paper has been improved further and hope it will receive favorable consideration for publication in PLOS ONE.

Thank you for the consideration of our manuscript

Sincerely,

Delvin Nyasani, on behalf of all authors

---

## [Editor Report · Decision Letter 1]

12 Jul 2023

PONE-D-23-02660R1Sexually transmitted infection knowledge levels, socio-demographic characteristics and sexual behaviour among men who have sex with men:  results from a cross-sectional survey in Nairobi, Kenya.PLOS ONE

Dear Dr. Nyasani,

Thank you for submitting your manuscript to PLOS ONE. After careful consideration, we feel that it has merit but does not fully meet PLOS ONE’s publication criteria as it currently stands. Therefore, we invite you to submit a revised version of the manuscript that addresses the points raised during the review process.

Dear Authors,Thanks so much for submitting the revised manuscript to PLOS ONE. Before the final decision lease consider and apply these comments. 1- As the main objective of this project was to study the knowledge about STI among MSM you clearly define this issue in the objective, please modify the title into "Sexually Transmitted Infection Knowledge among Men Who Have Sex with Men in Nairoubi Kenya" OR clearly add the other sections of the title into the objective. I recommend to define the objective of the study carefully based on the findings.2- For Bivariable analysis, please replace bivariable logistic regression instead of chi-2 or Fisher's Exact test and report crude OR.3- It is necessary to add those variables with a p_value <0.2 into the multivaraible and try to reduce non-significant tests based on backward elimination.  Please see Methods in Epidemiologic Research 2012. Online Free Available. 4- In Table 5, it is not clear the reported p_value is belong to bivariable or multivariable?5- Please add the limitations in one paragraph and remove them before the conclusion. Bes Regards============================

We look forward to receiving your revised manuscript.

Kind regards,

Hamid Sharifi

Academic Editor

PLOS ONE
---

## [Author Response · Author response to Decision Letter 1]

1 Aug 2023

UNIVERSITY OF NAIROBI

KAVI-INSTITUTE OF CLINICAL RESEARCH

Telephone ; 2717694, 2714613, 2725404

Mob : 0722 207 417 or 0734 333 143

Fax:2727703

Kenyatta National Hospital

P.O. Box 19676-00202

Nairobi, Kenya

Delvin Nyasani BSN 01 Aug 2023

Research Nurse

KAVI-Institute of Clinical Research

University of Nairobi

+254 720898197 (mobile)

Email: dnyasani@kaviuon.org

Hamid Sharifi

Academic Editor

Public Library of Science (PLOS ONE)

Dear Editor,

We are delighted that PLOS ONE will consider publication of our paper pending satisfactory revisions as suggested by the editor and reviewers. 

We have given careful consideration to all the editor’s comments and we have done our best to address them all. The following is a point by point explanation of how we have addressed the concerns and revised our manuscript. The line number(s) on the revised manuscript with Track Changes and text highlight has been specified to show the text representing each response. 

Editor’s comments

Thanks so much for submitting the revised manuscript to PLOS ONE. Before the final decision please consider and apply these comments. 

1- As the main objective of this project was to study the knowledge about STI among MSM you clearly define this issue in the objective, please modify the title into "Sexually Transmitted Infection Knowledge among Men Who Have Sex with Men in Nairoubi Kenya" OR clearly add the other sections of the title into the objective. I recommend to define the objective of the study carefully based on the findings.

Response

Thank you, we have modified our title. Kindly see line 1 and 2.Also, line 89-90 on the manuscript

2- For Bivariable analysis, please replace bivariable logistic regression instead of chi-2 or Fisher's Exact test and report crude OR.

Response

Thank you, we have revised the data analysis. Kindly see data analysis paragraph line 209-217 on the manuscript.

3- It is necessary to add those variables with a p_value <0.2 into the multivaraible and try to reduce non-significant tests based on backward elimination. Please see Methods in Epidemiologic Research 2012. Online Free Available. 

Response

Thank you, we have revised. Please see line 301-360. Also, table 5-8 on the manuscript.

4- In Table 5, it is not clear the reported p_value is belong to bivariable or multivariable?

Response

Thank you. We have revised. Please see line 301-360. Also, table 5-8 on the manuscript. 

5- Please add the limitations in one paragraph and remove them before the conclusion.

Response

Thank you for your suggestion. We have revised. Kindly see line 474-480 on the manuscript.

We thank the editor for comments. With these revisions, we feel the paper has been improved further and hope it will receive favorable consideration for publication in PLOS ONE.

Thank you for the consideration of our manuscript

Sincerely,

Delvin Nyasani, on behalf of all authors

---

## [Editor Report · Decision Letter 2]

2 Aug 2023

Sexually transmitted infection knowledge among men who have sex with men in Nairobi, Kenya.

PONE-D-23-02660R2

Dear Dr. Nyasani,

We’re pleased to inform you that your manuscript has been judged scientifically suitable for publication and will be formally accepted for publication once it meets all outstanding technical requirements.

Kind regards,

Hamid Sharifi

Academic Editor

PLOS ONE
---

## [Editor Report · Acceptance letter]

1 Sep 2023

PONE-D-23-02660R2 

Sexually transmitted infection knowledge among men who have sex with men in Nairobi, Kenya. 

Dear Dr. Nyasani:

I'm pleased to inform you that your manuscript has been deemed suitable for publication in PLOS ONE. Congratulations! Your manuscript is now with our production department. 

Kind regards, 

on behalf of

Dr. Hamid Sharifi 

Academic Editor

PLOS ONE